 RESEARCH ADVANCE

# Multiple Wnts act synergistically to induce Chk1/Grapes expression and mediate G2 arrest in *Drosophila* tracheoblasts

**Amrutha Kizhedathu[1,2], Rose Sebastian Kunnappallil[1†], Archit V Bagul[1‡], Puja Verma[1], Arjun Guha[1]***

[1]Institute for Stem Cell Science and Regenerative Medicine (inStem), GKVK Campus, Bangalore, India; [2]SASTRA University, Thirumalaisamudram, India

**Abstract** Larval tracheae of *Drosophila* harbour progenitors of the adult tracheal system (tracheoblasts). Thoracic tracheoblasts are arrested in the G2 phase of the cell cycle in an ATR (mei-41)-Checkpoint Kinase1 (grapes, Chk1) dependent manner prior to mitotic re-entry. Here we investigate developmental regulation of Chk1 activation. We report that Wnt signaling is high in tracheoblasts and this is necessary for high levels of activated (phosphorylated) Chk1. We find that canonical Wnt signaling facilitates this by transcriptional upregulation of Chk1 expression in cells that have ATR kinase activity. Wnt signaling is dependent on four Wnts (Wg, Wnt5, 6,10) that are expressed at high levels in arrested tracheoblasts and are downregulated at mitotic re-entry. Interestingly, none of the Wnts are dispensable and act synergistically to induce Chk1. Finally, we show that downregulation of Wnt signaling and Chk1 expression leads to mitotic re-entry and the concomitant upregulation of Dpp signaling, driving tracheoblast proliferation.

*For correspondence:
arjung@instem.res.in

Present address: †National Centre for Biological Sciences, Tata Institute of Fundamental Research, Bangalore, India; ‡Institute of Genetics and Molecular and Cellular Biology, Illkirch-Graffenstaden, France

Competing interests: The authors declare that no competing interests exist.

## Introduction

The development of a single cell into a multicellular organism is achieved by a highly regulated process of cell proliferation. Cells divide at the appropriate time and place or remain paused either in the G1/G0 (*Cheung and Rando, 2013*) or G2 (*Bouldin and Kimelman, 2014*) phases of the cell cycle. In this study, we investigate the mechanisms that regulate developmental G2 arrest.

The transition from G2 to M occurs via a mechanism that is conserved from yeast to humans. Mitotic entry is triggered by the activation of the Cyclin-dependant kinase 1 (Cdk1/Cdc2)-Cyclin B complex. Upon activation, the Cdc2-Cyclin B complex phosphorylates substrates in the cytoplasm and nucleus and prepares the cell for mitosis (*Bouldin and Kimelman, 2014*). The activation of the Cdc2-Cyclin B complex is dependent on the dephosphorylation of Cdc2 by the phosphatase Cdc25. Mechanisms regulating developmental G2 arrest have been studied extensively in *Drosophila* (*Johnston and Edgar, 1998*; *Ayeni et al., 2016*; *Otsuki and Brand, 2018*) and more recently in Zebrafish (*Nguyen et al., 2017*). An emerging theme is that cells paused in G2 lack the essential drivers of G2-M such as Cdc25 and Cyclin B respectively.

Our studies on G2 arrest have focused on the progenitors of the thoracic tracheal (respiratory) system of adult *Drosophila* (tracheoblasts). Tracheoblasts arrest in G2 for a period of ~56 hr during larval life and then enter mitosis at the onset of pupariation. The tracheoblasts, and the tracheae they comprise, grow in size while arrested (increase ~11 fold in volume) and undergo rapid size-reductive divisions thereafter (cell division time ~10 hr)(*Kizhedathu et al., 2018*). Contrary to other models of developmental G2 arrest in *Drosophila*, tracheoblasts express all the drivers necessary for G2-M and utilize a different mechanism for G2 arrest. We have shown previously that tracheoblasts co-opt the ATR (mei-41)/Chk1(Grapes) DNA damage checkpoint pathway for the induction of G2

arrest (*Kizhedathu et al., 2018*). Loss of ATR/Chk1 results in tracheoblasts initiating mitoses precociously.

What remained unclear in this study is the mechanism by which the ATR/Chk1 axis is controlled. Although the activation of ATR/Chk1 has typically been associated with DNA damage, there was no evidence for DNA damage in arrested tracheoblasts.

Developmental signals coordinate G2 arrest and G2-M by controlling the expression of positive regulators of G2-M. For example, a pulse of *Drosophila* steroid hormone Ecdysone has been shown to induce the expression of Cdc25/*Stg* mRNA in imaginal histoblasts (*Ninov et al., 2009*). Along these lines, the Notch and Wnt signaling pathways repress the expression of *Stg* in neural precursors in the wing imaginal disc (*Johnston and Edgar, 1998*). More recently, a study on neural stem cells has found that Insulin signaling upregulates Stg protein levels via repression of Tribbles expression; Tribbles is a kinase that targets Stg for proteosomal degradation (*Otsuki and Brand, 2018*). These studies led us to investigate the possibility that developmental signaling pathways control the activation of the ATR/Chk1 pathway in tracheoblasts.

To identify signals that regulate ATR/Chk1, we knocked down the components of each of ten developmental signaling pathways that regulate cell proliferation and growth and examined whether these perturbations result in precocious mitotic re-entry. We found that perturbations in Wnt signaling led to premature cell division akin to Chk1 mutants. This led us to examine the role of the Wnt signaling pathway in the regulation of the ATR/Chk1 pathway.

## Results

### Wnt signaling is required for G2 arrest in Tr2 tracheoblasts

The tracheal branches of the second thoracic metamere (Tr2) in the *Drosophila* larva are comprised of cells that also serve as the progenitors of the thoracic tracheal system of the adult animal. These tracheoblasts remain quiescent through most of larval life and rekindle a mitotic program prior to pupariation. Tracheoblasts in the Dorsal Trunk (DT) in Tr2 enter larval life in G1, transition from G1 to S to G2 in the first larval instar (L1) and remain arrested in G2 from second instar (L2) till 32–40 hr into third larval instar (L3) whereupon they re-enter mitosis (*Figure 1A*; *Kizhedathu et al., 2018*). These cells express high levels of Chk1 mRNA and phosphorylated Chk1 (pChk1) during the period in which they are paused in G2 and downregulate levels of Chk1 mRNA and pChk1 upon mitotic re-entry. Loss of Chk1 leads to precocious proliferation by 16–24 h L3,~24 hr earlier than expected (*Figure 1B*, *Figure 1—source data 1*).

To explore the possibility that developmental signals control the activation of the ATR/Chk1 axis in tracheoblasts, we knocked-down expression of essential transducers of the EGF, FGF, insulin/PI3K, Hedgehog, JAK/STAT, Ecdysone, JNK, Hippo, Wnt and Dpp signaling pathways in the trachea by RNA interference (under the control of trachea-specific *Breathless (btl)-GAL4* driver) and determined if these perturbations led to precocious mitotic re-entry at 16–24 hr in L3 (*Figure 1—figure supplement 1*). Of the abovementioned signaling pathways, downregulation of components of the Wnt pathway resulted in precocious proliferation at 16–24 h L3 (*Figure 1B*, *Figure 1—source data 1*). These findings led us to examine more carefully the role of Wnt signaling in the activation of the ATR/Chk1 axis in tracheoblasts.

The Wingless/Wnt signaling pathway regulates many different developmental processes in metazoans. Wnt genes encode secreted proteins that can act as signaling molecules and morphogens. The Wnt pathway is activated upon the binding of Wnt ligands to Frizzled (Fz) receptors. Ligand-receptor engagement leads to the activation of the cytoplasmic phosphoprotein Dishevelled (Dsh). Phosphorylated Dsh antagonizes an intracellular complex that targets β-Catenin for degradation. The consequent stabilization of β-Catenin facilitates its translocation to the nucleus where it binds the TCF/LEF family of transcription factors and initiates the transcription of downstream genes (*Bejsovec, 2006*). Apart from this 'canonical' mechanism, there are 'non-canonical' mechanisms for Wnt signaling that do not require β-Catenin (*Zhan et al., 2017*; *Swarup and Verheyen, 2012*).

To characterize the role of Wnt signaling in tracheoblasts, we knocked-down the abovementioned components of the pathway in the trachea by RNA interference and examined the timing of mitotic

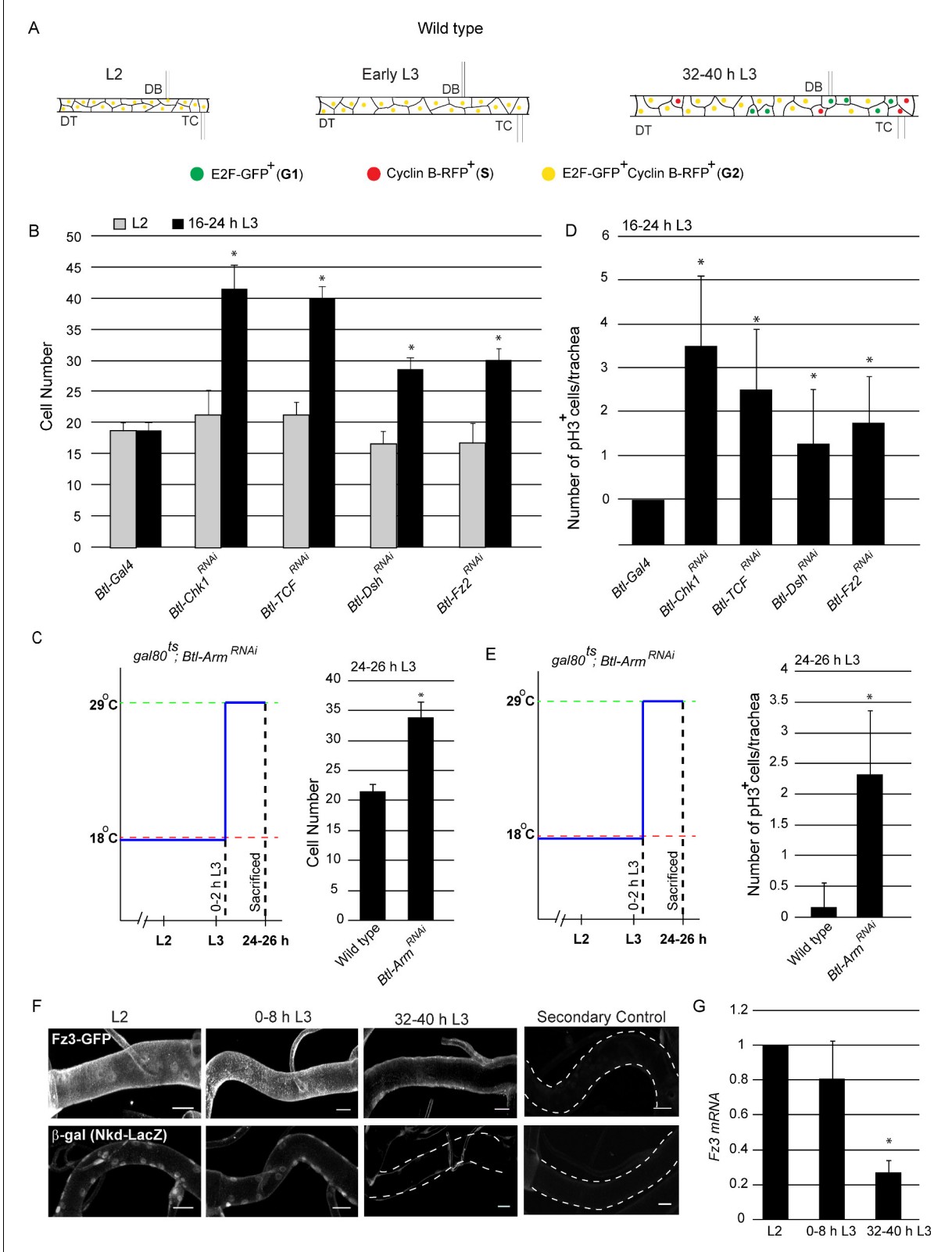

**Figure 1.** Wnt signaling is required for G2 arrest in Tr2 tracheoblasts. (**A**) Cartoon representing the cell cycle phasing of cells in wild type Tr2 DT at different larval stages based on FUCCI. (**B-C**) Impact of knockdown of *Chk1* and components of the Wnt signaling pathway on cell numbers in Tr2 DT. (**B**) Graph shows cells numbers in wild type (*btl-GAL4*), *Btl-Chk1*[RNAi] (*btl-GAL4/UAS-Chk1*[RNAi])*, and *Btl*-Wnt pathway components [RNAi] (*btl-GAL4/+; UAS-TCF*[RNAi]*/+, btl-GAL4/UAS-Dsh*[RNAi]*, btl-GAL4/UAS-Fz2*[RNAi]*) at L2 and 16–24 h L3. (**C**) Graph shows cell numbers in wild type (t*ub-GAL80*[ts]*/+; UAS-*

*Figure 1 continued on next page*

**Figure 1 continued**

*Arm^RNAi/Tb)* and Armadillo mutants (*tub-GAL80^ts/+; btl-GAL4/UAS-Arm^RNAi*) at 24–26 h L3. Larvae were grown at 18°C and transferred to 29°C at 0–2 hr into L3. (D-E) Impact of reduction in levels of *Chk1* or in levels of different components of the Wnt signaling pathway on mitotic indices in Tr2 DT. (D) Graph shows frequency of pH3⁺ nuclei at 16–24 h L3 in wild type (*btl-GAL4*), *Chk1* mutant (*btl-GAL4/UAS-Chk1^RNAi*), and Wnt pathway mutants (*btl-GAL4/+; UAS-TCF^RNAi/+, btl-GAL4/UAS-Dsh^RNAi, btl-GAL4/UAS-Fz2^RNAi*) (E) Graph shows the frequency of pH3⁺ nuclei in wild type (*tub-GAL80^ts/+; UAS-Arm^RNAi/Tb*) and Armadillo mutants (*tub-GAL80^ts/+; btl-GAL4/UAS-Arm^RNAi*) at 24–26 h L3. Larvae were grown as stated above. (F) Expression pattern of Wnt reporters Fz3-GFP (*fz3-GFP*) and Nkd- LacZ (*nkd-lacZ*) in Tr2 DT at different larval stages. Panels show immunostaining for GFP in Fz3-GFP (*fz3-GFP*) (top panel) and β-Gal in Nkd- LacZ (*nkd-lacZ*) (Bottom panel) and their respective secondary controls (G) Quantitative PCR analysis of *Fz3* mRNA levels in micro-dissected Tr2 DT fragments at different stages. Graph shows fold change in mRNA levels with respect to L2 (n = 3 experiments, n ≥ 15 Tr2 DT fragments/condition/experiment, mean ± standard deviation). DT = Dorsal Trunk, DB = Dorsal Branch, TC = Transverse Connective. Scale bar = 20 µm (mean ± standard deviation, n ≥ 7 tracheae) Student's paired t-test: *p<0.05.

The online version of this article includes the following source data and figure supplement(s) for figure 1:

**Source data 1.** *Figure 1B* Numerical data for number of cells in Tr2 DT of wild type, *Btl-Chk1^RNAi* and *Btl*-Wnt pathway RNAi larvae at different stages.
**Figure supplement 1.** List of developmental signaling pathway knockdowns not showing proliferation at 16–24 h L3.
**Figure supplement 2.** Loss of Chk1 or Wnt signaling leads to continuous proliferation of tracheoblasts through L3.

re-entry. To establish the timecourse of cell proliferation in the tracheae we utilized two assays. First, we counted the numbers of tracheoblasts in Tr2 DT at L2 and at 16–24 h L3. Second, we quantified the frequencies of phospho-histone H3⁺ (pH3⁺) mitotic figures. Knockdown of *Fz2, Dsh and TCF (Pangolin)* did not affect numbers of tracheoblasts in L2 but led to increased numbers at 16–24 h L3 (*Figure 1B,* n ≥ 7 tracheae per timepoint here and in all subsequent figures showing cell frequencies, *Figure 1—source data 1*). In our hands, the knockdown of *Drosophila* β-Catenin (*Armadillo (Arm)*) in the tracheal system from embryonic stages resulted in embryonic lethality. Thus, to characterize the role of β-Catenin in tracheoblasts at larval stages we knocked-down *Arm* expression in the trachea specifically at larval stages using the temperature sensitive Gal4-UAS-gal-80^ts system (see schematic in *Figure 1C*). Briefly, control (*tub-GAL80^ts/+; UAS-Arm^RNAi/Tb*) and *Btl-Arm^RNAi* (*tub-GAL80^ts/+; btl-GAL4/UAS-Arm^RNAi*) animals were grown at 18°C till 0–2 h L3 and moved to 29°C for 24 hr. The animals were dissected at 24–26 h L3 and the cell numbers in Tr2 DT were counted. Knockdown of *Arm* did not perturb numbers of cells in Tr2 DT in L2 but led to increased numbers of cells at 24 h L3 (*Figure 1C,* n ≥ 7 tracheae per condition, *Figure 1—source data 1*).

Apart from quantifying cell numbers at L2 and early L3, we assayed mitotic activity by immunostaining with an antibody against phosphorylated Histone H3 (pH3), a marker for cells undergoing mitosis. We found increased frequencies of pH3⁺ figures at early L3 in all backgrounds in which we observed increased numbers of cells at early L3 (*Figure 1D–E*). We also examined the timecourse of cell proliferation throughout L3 in wild type, Chk1 and Wnt pathway mutants. We found that tracheoblasts in Wnt pathway mutants, akin to Chk1 mutants, enter mitosis precociously and continue cell division thereafter (*Figure 1—figure supplement 2*). Together, the cell proliferation assays show that loss of Wnt signaling leads to precocious mitotic re-entry and proliferation.

Next, we investigated the spatiotemporal pattern of Wnt signaling in Tr2 DT during larval stages. For this, we examined the expression of two reporters for Wnt signaling (*Frizzled3 (Fz3)*-GFP and *Naked* (Nkd)-lacZ) at L2, 0–8 h L3 and 32–40 h L3 (*Sivasankaran et al., 2000; Xu et al., 2018; Tian et al., 2016*). Immunostaining using anti-GFP and anti-β-gal antisera respectively revealed that levels of expression of *fz3-GFP* and *nkd-lacZ* were high at L2 and 0–8 h L3 and significantly lower at 32–40 hr into L3 (*Figure 1F*, n = 6 tracheae per condition per experiment, n = 2). We also measured the expression of *Fz3* mRNA in Tr2 DT by quantitative real time PCR (qPCR). qPCR analysis showed that the expression of *Fz3* mRNA is high in L2 and early L3 and significantly lower at 32–40 h L3 (*Figure 1G*, n = 3). These data show that canonical Wnt signaling is active in G2 arrested cells and is downregulated upon mitotic re-entry.

## pChk1 levels are reduced in Wnt pathway mutants

Since perturbations in Wnt signaling led to precocious proliferation akin to the loss of Chk1, next we probed the role of Wnt pathway in Chk1 activation (phosphorylated Chk1, pChk1). pChk1 levels are high in tracheoblasts in L2 and early L3 and significantly lower at 32–40 h L3 (*Kizhedathu et al., 2018*). pChk1 immunostaining in *Btl-TCF^RNAi* at L2 showed that the levels of pChk1 were

dramatically reduced in comparison to wild type (*Figure 2A*, n = 6–8 tracheae per condition per experiment, n = 3). We inferred that the Wnt pathway is required Chk1 activation in the tracheoblasts.

## Wnt signaling upregulates Chk1 expression

Our earlier studies have shown that levels of pChk1 and Chk1 mRNA are correlated in Tr2 DT. In other words, both pChk1 and Chk1 mRNA levels are high in L2 and early L3 and diminish at 32–40 h L3. Studies in yeast (*Ford et al., 1994*), *Drosophila* (*Bayer et al., 2018*) and *Xenopus* (*Kumagai et al., 1998*) have shown that the overexpression of Chk1 is sufficient to induce G2 arrest. Thus, we hypothesized that Wnt signaling facilitates high levels of pChk1 via induction of high levels of Chk1 expression.

To investigate the relationship between Wnt signaling and Chk1 expression, we utilized two approaches. First, we quantified levels of Chk1 mRNA in Tr2 DT. Second, we characterized Chk1 promoter activity using an enhancer trap line (*Chk1-lacZ*). We quantified the levels of Chk1 mRNA in microdissected Tr2 fragments from wild type, *Btl-TCF^RNAi* and *Btl-ArmS10 (*an activated form of *Armadillo* (*Btl-ArmS10*)) at different larval stages using quantitative real-time PCR (qPCR). The loss of Wnt signaling (*Btl-TCF^RNAi*) led to a reduction in *Chk1* levels at L2 and early L3; the levels of expression at these stages were comparable to 32–40 h L3. Conversely, overexpression of *ArmS10* led to an increase in the level of Chk1 transcript (*Figure 3A*, ≥15 tracheal fragments per timepoint per experiment, n = 3 experiments). Next, we examined the expression of *Chk1-lacZ*. Anti-β-gal immunostaining of tracheae from *Chk1-lacZ* animals at different larval stages showed that β-gal expression is high in L2 and early L3 and declines at 32–40 hr into L3 (*Figure 3B*, n = 6 tracheae per condition per experiment, n = 2). We then analysed the expression of the reporter in *Btl-TCF^RNAi* animals at L2. We found that β-gal expression was considerably diminished in L2 (*Figure 3B*, n = 6 tracheae per condition per experiment, n = 2). Together, the analysis of Chk1 mRNA levels and *Chk1-lacZ* in control and Wnt signaling-deficient tracheoblasts show that Wnt signaling is required for high levels of *Chk1* expression in G2-arrested cells.

To substantiate our findings that Wnt signaling regulates G2 arrest via control of Chk1 expression, we overexpressed *Chk1* or *ATR* in *Btl-TCF^RNAi* animals and determined if this prevented precocious proliferation and restored levels of pChk1. Cell counts in Tr2 DT at 16–24 h L3 in Wild type,

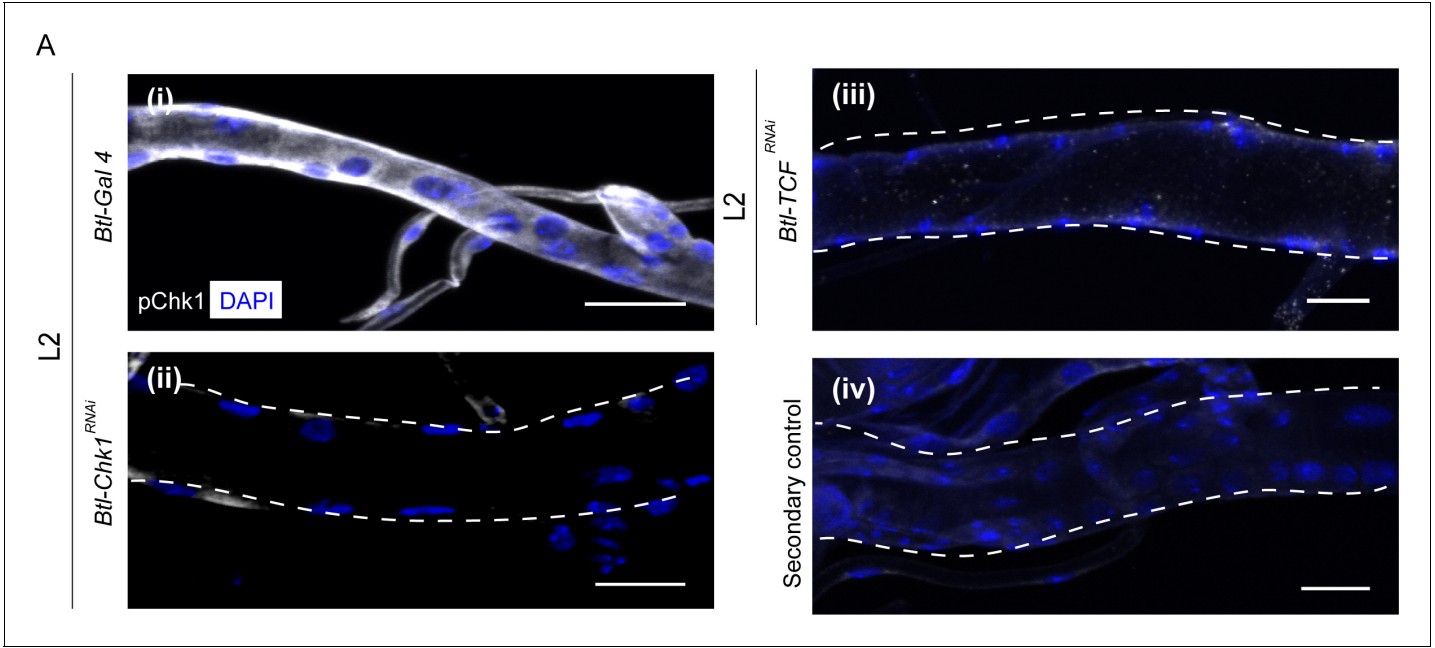

**Figure 2.** Phosphorylated Chk1 levels are diminished in the absence of Wnt signaling. (A) Activated Chk1 (phospho-Chk1^Ser345, pChk1) immunostaining (white) in Tr2 DT in Chk1 mutants and Wnt pathway mutants. pChk1 immunostaining in Tr2 DT in (i) wild type (*btl-GAL4*), (ii) *Btl-Chk1^RNAi (btl-GAL4/ UAS-Chk1^RNAi*) (iii) *Btl-TCF^RNAi (btl-GAL4/+; UAS-TCF^RNAi/+*) at L2 and (iv) wild type treated with secondary antibody alone. Scale bar = 20 μm.

*Btl-TCF^{RNAi}*, *Btl-TCF^{RNAi}*, *Chk1* (*TCF* knockdown with *Chk1* overexpression) and *Btl-TCF^{RNAi}*, *ATR* (*TCF* knockdown with *ATR* overexpression) expressing animals showed that Chk1 overexpression did indeed prevent premature mitotic re-entry (*Figure 3C*, *Figure 3—source data 1*). The overexpression of ATR did not (*Figure 3C*, *Figure 3—source data 1*). Immunostaining for pChk1 revealed that the levels were high and comparable to wild type in *Btl-TCF^{RNAi}*, *Chk1* animals. In contrast, the levels

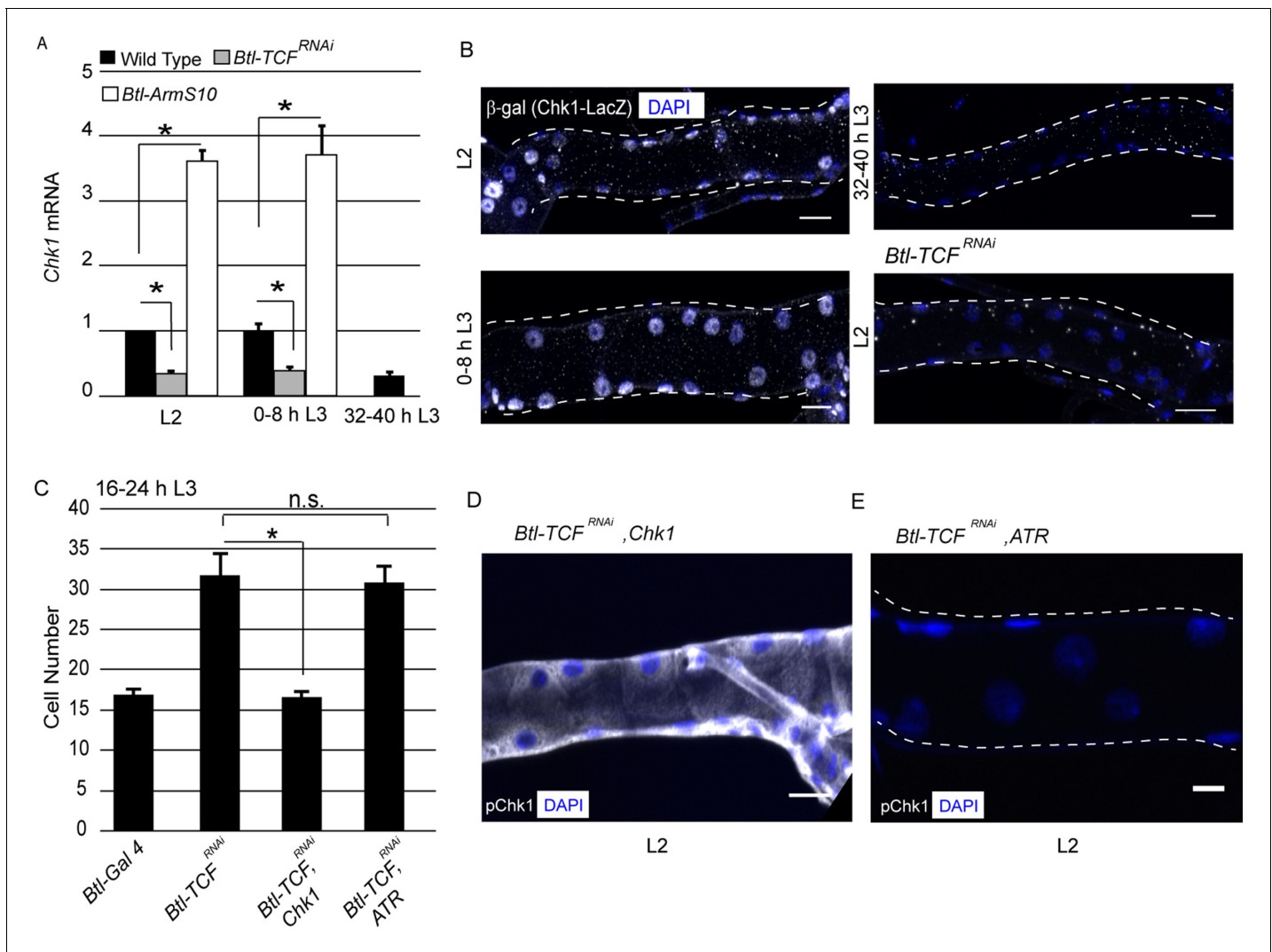

**Figure 3.** Wnt signaling regulates Chk1 transcription. (A) Quantitative PCR analysis of *Chk1* mRNA levels in micro-dissected Tr2 DT fragments at different stages. Graph shows fold change in *Chk1* mRNA in Tr2 DT fragments from wild type (*btl-GAL4*), Wnt pathway loss-of-function (*btl-GAL4/+; UAS-TCF^{RNAi}/+*) and Wnt pathway gain-of-function (*UAS-ArmS10/+; btl-GAL4/+*) larvae. Fold change has been represented with respect to L2 (n = 3 experiments, n ≥ 15 Tr2 DT fragments/condition/experiment, mean ± standard deviation). (B) Analysis of *Chk1*-lacZ expression in Tr2 DT in wild type and Wnt pathway mutants. β-Gal immunostaining in larvae expressing *Chk1-lacZ* at L2, 0–8 h L3 and 32–40 h L3 and in *Btl-TCF^{RNAi}* (*btl-GAL4/Chk1 lacZ; UAS-TCF^{RNAi}/+*) at L2. (C) Impact of overexpression of *Chk1* and *ATR* in TCF mutants. Graph shows cell numbers in wild type (*btl-GAL4, UAS-FUCCI/ Cyo GFP; +/Tb*), *Btl-TCF^{RNAi}* (*btl-GAL4, UAS-FUCCI/+; UAS-TCF^{RNAi}/+*), *Btl-TCF^{RNAi}*, *Chk1* (*btl-GAL4, UAS-FUCCI/+; UAS-TCF^{RNAi}/UAS-Chk1*) and *Btl-TCF^{RNAi}*, *ATR* (*btl-GAL4, UAS-FUCCI/+; UAS-TCF^{RNAi}/UAS ATR*) at 16–24 h L3. (D) Impact of reduction of *TCF* and overexpression of *Chk1* on levels of pChk1 in Tr2 DT. pChk1 immunostaining (white) in *Btl-TCF^{RNAi}*, *UAS-Chk1* (*btl-GAL4, UAS-FUCCI/+; UAS-TCF^{RNAi}/UAS-Chk1*) animals at L2 (This image has been acquired at a lower laser power and gain setting compared to *Figure 2A* to prevent saturation of white pixels). (E) Impact of reduction of *TCF* and overexpression of *ATR* on levels of pChk1 in Tr2 DT. pChk1 immunostaining (white) in *Btl-TCF^{RNAi}*, *UAS-ATR* (*btl-GAL4, UAS-FUCCI/+; UAS-TCF^{RNAi}/UAS-ATR*) animals at L2 (mean values ± standard deviation, n ≥ 7 tracheae) Scale bar = 20 μm. Student's paired t-test: *p<0.05.

The online version of this article includes the following source data for figure 3:

**Source data 1.** *Figure 3C* Numerical data for number of cells in Tr2 DT of wild type, *Btl-TCF^{RNAi}*, *Btl-TCF^{RNAi}*, *Chk1* and *Btl-TCF^{RNAi}*, *ATR* expressing larvae at 16–24 h L3.

of pChk1 were not restored in *Btl-TCF^RNAi*, *ATR* expressing animals and were comparable to *Btl-Chk1^RNAi* animals at the same stage (*Figure 3D–E*, n = 6 tracheae per experiment, n = 2). We independently confirmed that the overexpression of ATR did result in increased levels of ATR mRNA expression by qPCR (data not shown). These data support the hypothesis that Wnt dependent regulation of Chk1 expression is necessary for G2 arrest in Tr2 DT.

## *Wg, Wnt5, Wnt6* and *Wnt10* are required for Chk1 expression

Having identified Wnt signaling as a regulator of Chk1 expression and G2 arrest, next we focused on our attention on the mechanism by which the pathway is activated in the tracheal system. Since Wnt signaling is high when the cells are arrested and diminished once cells start dividing, we surmised that the ligands that activate Wnt signaling might also be expressed in the tracheal system in the same temporal pattern. To investigate this possibility, we measured levels of various Wnt ligand mRNAs in microdissected Tr2 fragments using qPCR. The analysis revealed that mRNA levels of 4 Wnt ligands – *Wg, Wnt5, Wnt6* and *Wnt10* - are high in L2 and 0–8 h L3 and low at 32–40 h L3 (*Figure 4A*, ≥15 tracheal fragments per timepoint per experiment, n = 3 experiments). *Wnt2, Wnt4* and *WntD* were undetectable in the tracheae at all stages (data not shown). The qPCR data showed that the timecourse of expression of some Wnt ligands in Tr2 DT paralleled the timecourse of Wnt signaling in this segment.

To corroborate the qPCR data and examine the spatial pattern of Wnt ligand expression, we performed single molecule FISH (smFISH) for *Wg, Wnt5, Wnt6* and *Wnt10*. To confirm that the technique was working optimally in our hands, we first examined the distribution of *GAPDH*-specific probes. As expected, smFISH signals were detected in all tracheal cells at L2 and L3. We detected both prominent nuclear spots, representing putative sites for mRNA transcription, and faint spots in the cytoplasm (*Figure 4—figure supplement 1, C*). Next we examined the expression of the Wnt ligands. smFISH for *Wg, Wnt5, Wnt6* and *Wnt10* detected both nuclear spots and faint spots in the cytoplasm in virtually all the cells in Tr2 DT at L2 (*Figure 4B*) and 0–8 h L3 (data not shown). Neither the prominent nuclear spots (*Figure 4B* arrowheads) nor faint cytoplasmic signals were detectable at 32–40 h L3. To confirm that smFISH was indeed ligand-specific, we hybridized probes for *Wnt5* to trachea from *Wnt5[400]* null mutants (*Fradkin et al., 2004*). We did not detect any specific signal in these tracheae at L2 (*Figure 4—figure supplement 1, B*). Together, the qPCR and smFISH data show that the timecourse of expression of *Wg, Wnt5, Wnt6* and *Wnt10* in tracheoblasts in Tr2 DT parallels the timecourse of Wnt signaling in this segment and implicates these ligands as the activators of the Wnt pathway.

Next we examined how perturbations in the expression of the abovementioned ligands affect the timing of mitotic entry in Tr2 DT. Here again, we quantified numbers of cells in Tr2 DT at L2 and 16–24 h L3 and frequencies of pH3^+ nuclei in Tr2 DT at 16–24 h L3. Knockdown of *Wg, Wnt5, Wnt6* and *Wnt10* in the trachea led to an increase in cell number at 16–24 hr into L3 (*Figure 4C*, *Figure 4—source data 1*). There was also an increase in pH3^+figures in these backgrounds at 16–24 h L3 (*Figure 4D*). To validate these observations, we quantified the cell numbers in *Wg (Wg^ts)* (*Swarup and Verheyen, 2012*), *Wnt5 (Wnt5[400])* (*Fradkin et al., 2004*) and *Wnt6(Wnt6 KO)* (*Doumpas et al., 2013*) mutants and in a different RNAi line for *Wnt10*. *Wg^ts* animals were grown at 18°C till 0–2 h L3 and moved to 29°C for 24 hr. The animals were then dissected at 24–26 h L3 and the cell numbers in Tr2 DT were counted. We found an increase in cell numbers in *Wg^ts* animals that were raised at 29°C (*Figure 4E*, *Figure 4—source data 1*). Cell counts at L2 and16-24 h L3 in *Wnt5[400], Wnt6* mutants and *Btl-Wnt10^RNAi (btl-GAL4/UAS-Wnt10^RNAi)* showed that there were increased numbers of cells in these backgrounds at 16–24 h L3 (*Figure 4E*, *Figure 4—figure supplement 1A*, *Figure 4—source data 1*). Taken together, these data show that *Wg, Wnt5, Wnt6* and *Wnt10* are required to maintain G2 arrest in Tr2 DT. We also quantified cell numbers upon knockdown of the Wnt ligands that are not detected (*Wnt2, Wnt4* and *WntD*) and there was no increase in cell numbers at 16–24 h L3 in these backgrounds (*Figure 4—figure supplement 2*).

Since *Wg, Wnt5, Wnt6* and *Wnt10* are high in L2 and early L3 and the loss of any one of these ligands results in precocious proliferation, we examined whether the ligands could impact expression of each other. To test whether the knockdown of one of the ligands impacts the expression of others we measured mRNA levels of *Wg, Wnt5, Wnt6* and *Wnt10* in microdissected Tr2 DT fragments from *Btl-Wg^RNAi*, *Btl-Wnt5^RNAi*, *Btl-Wnt6^RNAi*, and *Btl-Wnt10^RNAi* animals respectively. We found that the reduction of expression of any one ligand did not impact the expression of others (*Figure 4—figure*

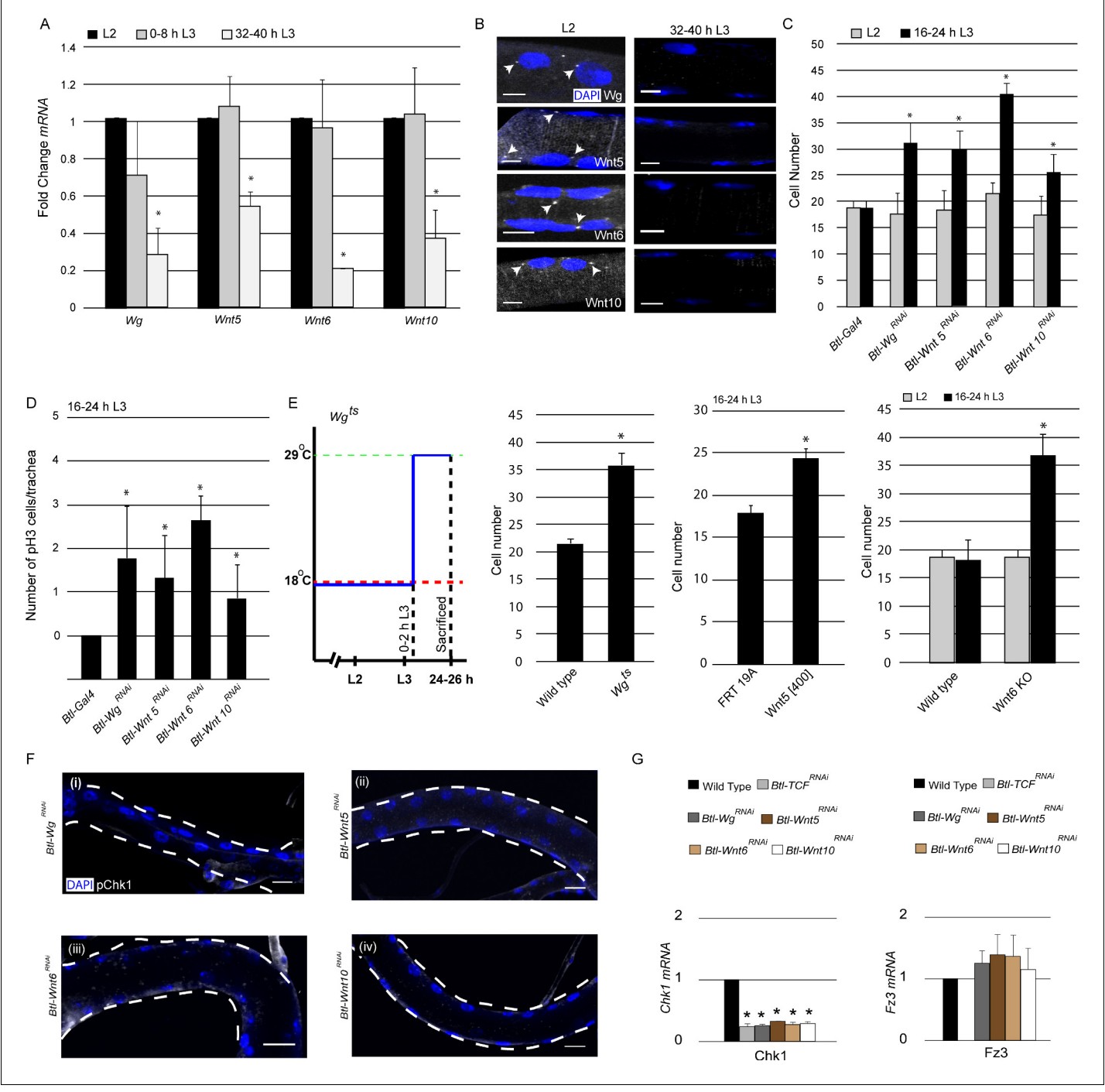

**Figure 4.** Wg, Wnt5, Wnt6 and Wnt10 are required to maintain Chk1 expression. (**A**) Quantitative PCR analysis for levels of *Wg, Wnt5, Wnt6 and Wnt10* mRNA in micro-dissected Tr2 DT fragments at different stages. Graph shows fold change in *Wg, Wnt5, Wnt6 and Wnt10* mRNA in Tr2 DT fragments from wild type (*btl-GAL4)* larvae at L2, 0–8 h L3 and 32–40 h L3. Fold change has been represented with respect to L2 (n = 3 experiments, n ≥ 15 Tr2 DT fragments/condition/experiment, mean ± standard deviation). (**B**) smFISH for *Wg, Wnt5, Wnt6 and Wnt10* mRNA in Tr2 DT at L2 and 32–40 h L3. Arrowheads point to the sites of mRNA accumulation. (Scale bar = 5 μm) (**C**) Impact of knockdown of components of the Wnt signaling pathway on cell numbers in Tr2 DT. Graph shows cells numbers in wild type (*btl-GAL4*, Same as *Figure 1B*) and *Btl*-Wnt pathway components[RNAi] (*btl-GAL4/+; UAS-Wg[RNAi]/+, btl-GAL4/+; UAS-Wnt5[RNAi]/+ btl-GAL4/+; UAS-Wnt6[RNAi]/+, btl-GAL4/+; UAS-Wnt10[RNAi]/+*) at L2 and 16–24 h L3. (**D**) Impact of reduction in levels of different components of the Wnt signaling pathway on mitotic indices in Tr2 DT. Graph shows frequency of pH3[+] nuclei at 16–24 h L3 in wild type (*btl-GAL4*, Same as *Figure 1D*) and Wnt pathway mutants (*btl-GAL4/+; UAS-Wg[RNAi]/+, btl-GAL4/+; UAS-Wnt5[RNAi]/+, btl-GAL4/+; UAS-Wnt6[RNAi]/+, btl-GAL4/+; UAS-Wnt10[RNAi]/+*) (**E**) Impact of expression of mutant *Wg, Wnt5 and Wnt6* on cell numbers in Tr2 DT. Graphs show cell numbers in wild type and *Wg* mutants (*Wg[ts]*) at 24–26 h L3, (Larvae were grown at 18°C and transferred to 29°C at 0–2 hr into L3) *Wnt5* mutant (*Wnt5*

*Figure 4 continued on next page*

Figure 4 continued

[400] FRT19A) and *Wnt6 (Wnt6 KO)* on cell numbers in Tr2 DT. (F) Activated Chk1 (phospho-Chk1$^{Ser345}$, pChk1) immunostaining (white) in Tr2 DT in Wnt pathway mutants. pChk1 immunostaining in Tr2 DT in (i) *Btl-Wg$^{RNAi}$*(*btl-GAL4/+; UAS-Wg$^{RNAi}$/+*), (ii) *Btl-Wnt5$^{RNAi}$*(*btl-GAL4/+; UAS-Wnt5$^{RNAi}$/+*), (iii) *Btl-Wnt6$^{RNAi}$*(*btl-GAL4/+; UAS-Wnt6$^{RNAi}$/+*) and *Btl-Wnt10$^{RNAi}$*(*btl-GAL4/+; UAS-Wnt10$^{RNAi}$/+*) at L2. (G) Quantitative PCR analysis of *Chk1 and Fz3* mRNA levels in micro-dissected Tr2 DT fragments at L2. Graphs show fold change in *Chk1 and Fz3* mRNA in Wild type, *Btl-TCF$^{RNAi}$*(*btl-GAL4/+; UAS-TCF$^{RNAi}$/+*), *Btl-Wg$^{RNAi}$* (*btl-GAL4/+; UAS-Wg$^{RNAi}$/+*), *Btl-Wnt5$^{RNAi}$*(*btl-GAL4/+; UAS-Wnt5$^{RNAi}$/+*), *Btl-Wnt6$^{RNAi}$*(*btl-GAL4/+; UAS-Wnt6$^{RNAi}$/+*) and *Btl-Wnt10$^{RNAi}$*(*btl-GAL4/+; UAS-Wnt10$^{RNAi}$/+*) in Tr2 DT fragments. Fold change has been represented with respect to Wild type (n = 3 experiments, n ≥ 15 Tr2 DT fragments/condition/experiment, mean ± standard deviation).(mean values ± standard deviation, n ≥ 7 tracheae) Scale bar = 20 μm. Student's paired t-test: *p<0.05.

The online version of this article includes the following source data and figure supplement(s) for figure 4:

**Source data 1.** *Figure 4C* Numerical data for number of cells in Tr2 DT of wild type, *Btl-Wg$^{RNAi}$*, *Btl-Wnt5$^{RNAi}$*, *Btl-Wnt6$^{RNAi}$* and *Btl-Wnt10$^{RNAi}$* larvae at different stages.
**Figure supplement 1.** Wnt ligands do not affect each others expression.
**Figure supplement 2.** Loss of *Wnt2, Wnt4* or *WntD* does not lead to precocious proliferation in Tr2 DT.

*supplement 1A*,≥15 tracheal fragments per timepoint per experiment, n = 3 experiments). Based on this analysis, we inferred that the expression of each of the Wnt ligands is independent of the others and that each ligand contributes toward G2 arrest.

Next we probed whether perturbations in *Wg, Wnt5, Wnt6* and *Wnt10* expression affected levels of Chk1 (pChk1 and *Chk1* mRNA). pChk1 immunostaining of *Btl-Wg$^{RNAi}$*, *Btl-Wnt5$^{RNAi}$*, *Btl-Wnt6$^{RNAi}$* and *Btl-Wnt10$^{RNAi}$* showed that levels of pChk1 in each of these backgrounds was diminished, the extent of the reduction in staining was comparable to what was observed in *Btl-Chk1$^{RNAi}$* expressing animals (*Figure 4F*, compare with *Figure 2A* (ii)). Quantification of *Chk1* mRNA in *Btl-TCF$^{RNAi}$*, *Btl-Wg$^{RNAi}$*, *Btl-Wnt5$^{RNAi}$*, *Btl-Wnt6$^{RNAi}$* and *Btl-Wnt10$^{RNAi}$* showed that the loss of any one of the ligands led to diminished levels of Chk1 mRNA expression. The levels of mRNA in each of these mutant backgrounds was comparable to the levels detected in *Btl-TCF$^{RNAi}$* (*Figure 4G*). Together, these data showed that all four ligands regulate Chk1 expression in Tr2 DT. In parallel with the analysis of Chk1 mRNA, we also examined the levels of *Fz3* mRNA, a conventional target of Wnt signaling. There was no significant change in *Fz3* mRNA levels in *Btl-Wg$^{RNAi}$*, *Btl-Wnt5$^{RNAi}$*, *Btl-Wnt6$^{RNAi}$* and *Btl-Wnt10$^{RNAi}$* (*Figure 4G*). However, the levels of *Fz3* were considerably reduced in *Btl-TCF$^{RNAi}$* We infer that the expression of any one of the abovementioned Wnt ligands is sufficient to induce *Fz3* expression in Tr2 DT but that all four are required to induce Chk1 expression.

## Exit from G2 arrest is required for activation of mitogenic signaling

Our analysis shows that loss of Wnt signaling and Chk1 expression leads to precocious mitotic re-entry in early L3. The subsequent timecourse of cell proliferation in these mutants reveals that tracheoblasts that enter mitosis prematurely in L3 continue to divide post mitotic re-entry (*Figure 1—figure supplement 2*). The reason that Wnt/Chk1-deficient tracheoblasts continue to divide after exiting G2 arrest was unclear to us. A previous study has identified Dpp (the *Drosophila* homolog of TGF-β) as the signal required for proliferation of tracheoblasts post mitotic re-entry. Dpp signaling is initiated by the binding of the ligand (Dpp) to the receptor Serine/Threonine Kinase Thickveins (Tkv). This in turn leads to the phosphorylation of the transcriptional activator Mad. Phosphorylated Mad (pMad) translocates to the nucleus and regulates gene expression. Pertinently, an earlier study on tracheoblasts showed that Dpp signaling (pMad immunostaining) was undetectable in cells in G2 arrest and observed mid L3 onwards once the cells start dividing (*Djabrayan and Casanova, 2016*). In light of the proliferation in Chk1 and Wnt pathway mutants, we examined the levels of Dpp signaling in these mutant backgrounds prior to and post mitotic re-entry.

As reported previously, pMad immunostaining was undetectable in L2 and early L3 (0–8 h L3 and 16–24 h L3) and nuclear localized at 32–40 h L3, once the cells entered mitosis (*Figure 5A*). We stained *Btl-Chk1$^{RNAi}$* and *Btl-Wnt6$^{RNAi}$* at 16–24 h L3 for pMad and detected abundant nuclear localized signal (*Figure 5B*). This suggested that release from G2 arrest leads to the concomitant upregulation of Dpp signaling that further stimulates proliferation. Next we investigated how the expression of an activated form of *Tkv (Btl-Tkv$^{QD}$)* (*Djabrayan and Casanova, 2016*) impacts pMad levels in arrested cells. We found that pMad immunostaining was undetectable in *Btl-Tkv$^{QD}$* animals at L2 (*Figure 5C*). Taken together, these observations suggest that arrest in G2 prevents the

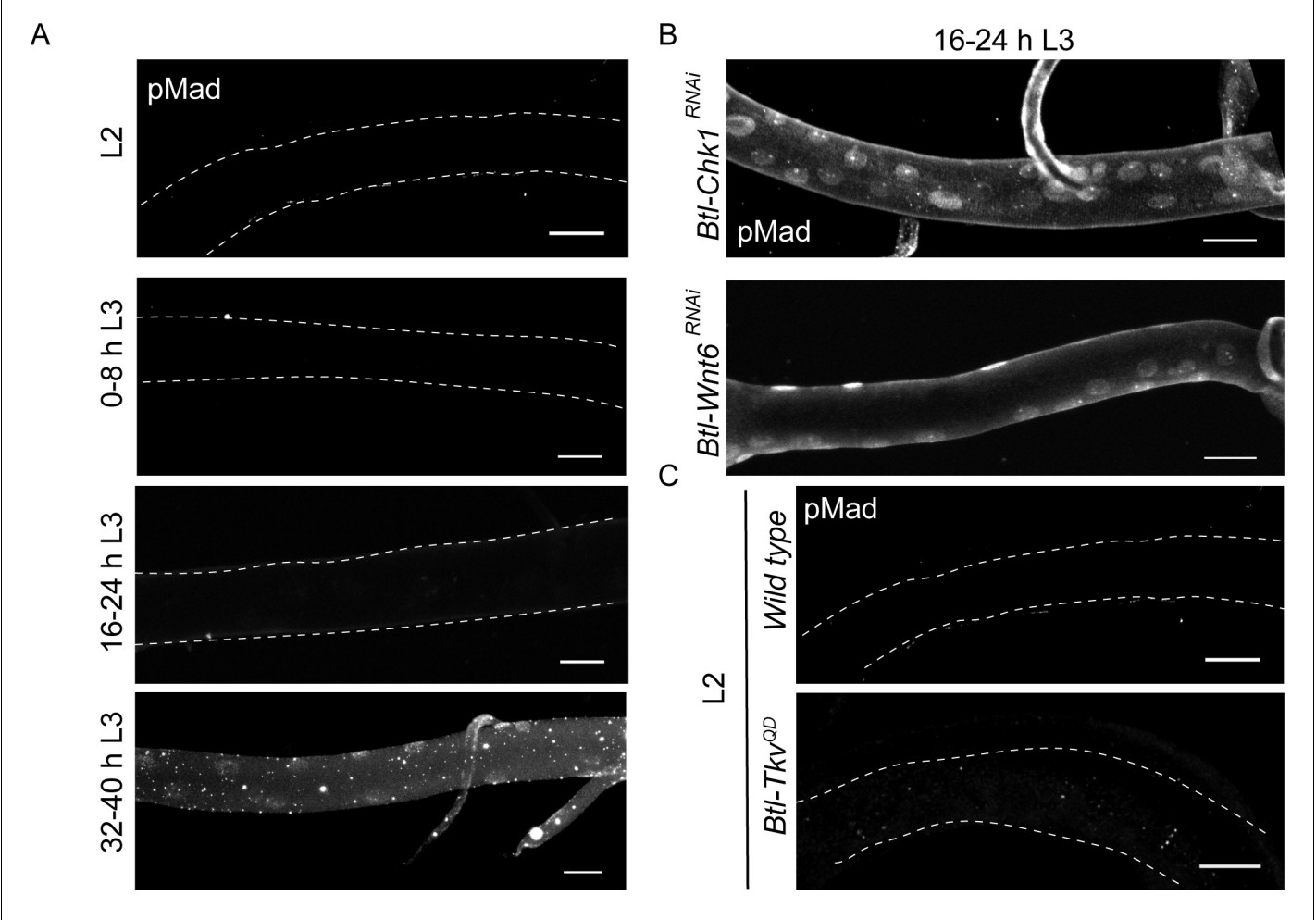

**Figure 5.** Exit from G2 arrest is required for activation of Dpp signaling. (**A**) phospho-Mad (pMad) immunostaining in wild type Tr2 DT at L2, 0–8 h L3, 16–24 h L3 and 32–40 h L3. (**B**) Impact of reduction of *Chk1* or Wnt pathway components on pMad expression in Tr2 DT. pMad immunostaining in Tr2 DT from *Btl-Chk1$^{RNAi}$ (btl-GAL4/UAS-Chk1$^{RNAi}$)* and *Btl-Wnt6$^{RNAi}$ (btl-GAL4/+; UAS-Wnt6$^{RNAi}$/+)* animals at 16–24 h L3. (**C**) Impact of overexpression of activated form of Tkv (*Btl-Tkv$^{QD}$*) on pMad expression in Tr2 DT. pMad immunostaining in Tr2 DT from wild type and *Btl-Tkv$^{QD}$ (btl-GAL4/+; UAS-Tkv$^{QD}$ /+)* animals at L2. Scale bar = 20 µm.

activation of Dpp signaling and that exit from G2 leads to the activation of Dpp signaling and continued cell proliferation.

## Juxtaposition of positive and negative regulators is required for optimal growth of Tr2 DT

The mechanism for G2 arrest in tracheoblasts is integral to their growth and to the growth of the tracheae they comprise. We have previously reported that arrested tracheoblasts, and the tracheae they comprise, grow significantly during the period in which the cells are arrested in G2. Premature exit from arrest resulting from the knockdown of Chk1 retards growth and results in a reduction in the overall size of Tr2 DT (*Figure 6A*, *Kizhedathu et al., 2018*). In light of the role of Wnt signaling in the regulation of *Chk1*, we examined how Wnt mutants impact the growth of the tracheae. Quantification of percentage growth between L2 and 32–40 h L3 revealed that compared to a 268 ± 32% (n = 6 tracheae) growth in length and 247 ± 19% (n = 12 tracheae) growth in width in control animals (*Kizhedathu et al., 2018*), TCF mutants grew only by 178 ± 8% (n = 6 tracheae) in length and 176 ± 13% (n = 6 tracheae) in width. Thus, consistent with our earlier finding that Chk1 is required for optimal tracheal growth, Wnt signaling is also required for growth of the thoracic tracheae during larval stages.

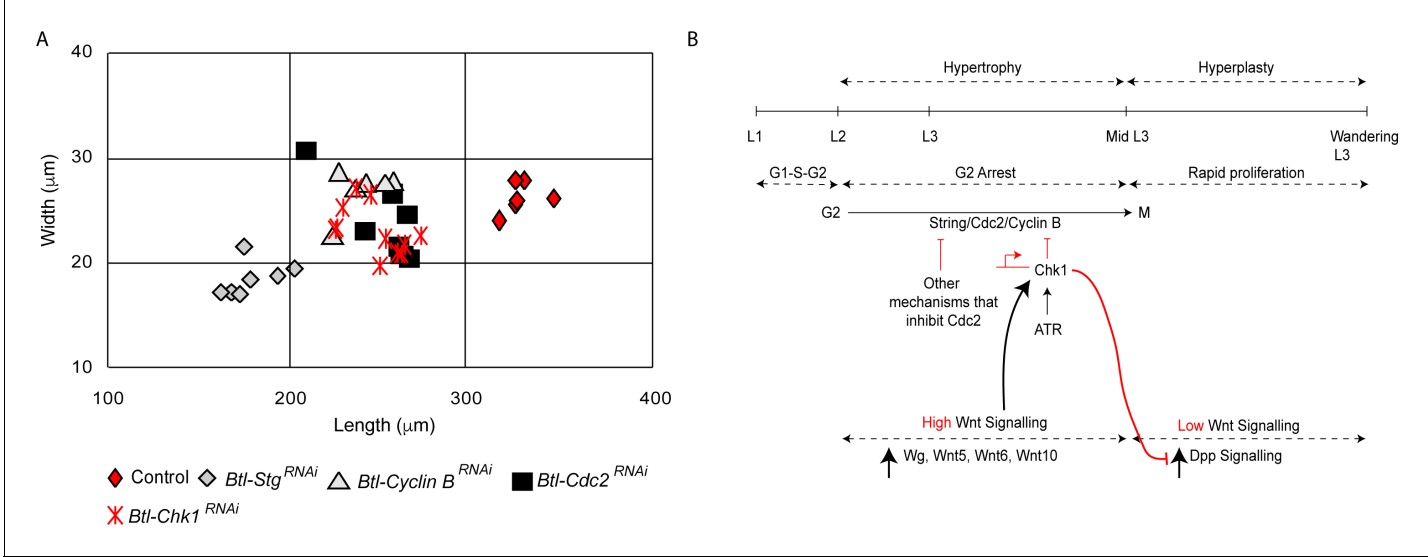

**Figure 6.** Loss of Stg, Cdc2 or Cyclin B perturbs hypertrophic growth of Tr2 DT. (**A**) Impact of reduction of Chk1 or different drivers of G2-M on the size of Tr2 DT at 32–40 h L3. Scatter plot shows length and width of Tr2 DT from Wild type, *Btl-Chk1^RNAi (btl-GAL4/UAS-Chk1^RNAi)*, *Btl-Stg^RNAi (btl-GAL4/+; UAS-Stg^RNAi/+)*, *Btl-Cdc2^RNAi(btl-GAL4/UAS-Cdc2^RNAi)* and *Btl-CyclinB^RNAi (btl-GAL4/+; UAS-CyclinB^RNAi/+)* at 32–40 h L3. Values shown in red are from data previously published in *Kizhedathu et al., 2018*. (n ≥ 6 tracheae). (**B**) Model for the regulation of G2 arrest by Wnt signaling. Wnt signaling negatively regulates G2-M by transcriptionally upregulating Chk1. Arrest in G2 negatively regulates Dpp signaling, preventing precocious proliferation and allowing for hypertrophic growth of Tr2 DT.

An unusual aspect of G2 arrest in Tr2 DT is that the cells express both drivers of G2-M (Cdc2/Cdk1, CyclinB, Stg) and simultaneously activate negative regulators of G2-M (Chk1, [*Kizhedathu et al., 2018*], this study). The significance of Cdc2, CycB and Stg expression in arrested tracheoblasts has been unclear. We hypothesized that the expression of positive regulators of G2-M could be necessary for growth. To test this hypothesis, we knocked down *Stg, Cdc2/Cdk1* and *Cyclin B* in the trachea and measured the length and width of Tr2 DT at 32–40 h L3. The loss of any one of these G2-M drivers led to a reduction in the size of Tr2 DT (*Figure 6A*). These data suggest that the juxtaposition of positive and negative regulators is required for the growth of the tracheoblasts and the tracheae they comprise (see model in *Figure 6B*).

## Discussion

Progenitors of the thoracic tracheal system of adult *Drosophila* arrest in G2 in a Chk1-dependent manner during larval stages. In this study we have investigated the mechanism for Chk1 activation in tracheoblasts. We find that Chk1 is transcriptionally upregulated in arrested tracheoblasts by the canonical Wnt signaling pathway and that this upregulation is sufficient to induce arrest. Our studies reveal that Wnt signaling is activated by four Wnt ligands - *Wg, Wnt5, Wnt6, Wnt10* – that are all expressed in the tracheoblasts. We show that all four ligands are required for Chk1 upregulation and that the temporal expression of the ligands is also under transcriptional control. *Wg, Wnt5, Wnt6,* and *Wnt10* are high when the cells are arrested in G2 and low at the time the cells enter division.

A fascinating aspect of Wnt signaling in the trachea is that as many as four ligands appear to be required for G2 arrest; *Wg, Wnt5, Wnt6* and *Wnt10* are all necessary. The loss of any one of the ligands leads to a reduction in Chk1 mRNA levels in tracheoblasts comparable with the knockdown of TCF. Interestingly, our data also show that these 4 Wnts act redundantly to regulate *Fz3* in the same cells. We hypothesize that the ligands act synergistically to maintain a threshold of Wnt signaling necessary for *Chk1* expression. Consistent with this, we found that overexpression of *Wnt5* or *Wnt10* in *Wnt6^RNAi*-expressing animals was able to prevent precocious proliferation at 16–24 h L3 (*Btl-Wnt6^RNAi* at 16–24 h L3 Avg: 32.4 ± 1.2 cells, n = 8 tracheae; *Btl-Wnt6^RNAi, Wnt5* at 16–24 h L3

Avg: 19.1 ± 0.8, n = 8 tracheae; *Btl-Wnt6*$^{RNAi}$, *Wnt10* at 16–24 h L3 Avg: 18.2 ± 1.7 cells, n = 8 tracheae).

It is noteworthy that *Wnt5*, which in *Drosophila* has been implicated in non-canonical Wnt signaling (*Shimizu et al., 2011*), facilitates canonical Wnt signaling in tracheoblasts. There is evidence in other systems that non-canonical Wnt ligands can activate canonical Wnt signaling (*Fu et al., 2016*; *Lyons et al., 2004*). Preliminary expression profiling of the larval tracheal system has indicated that both Derailed and Doughnut, RYK family receptors that mediate non-canonical Wnt signaling, are expressed (AG, AK unpublished). This suggests that Wnt5 could act either directly or indirectly to facilitate canonical Wnt signaling in tracheoblasts.

The role of the Wnt pathway in activation of the ATR/Chk1 axis in a developmental context has, to the best of our knowledge, not been reported previously. However, there is evidence that Wnt signaling can induce the DNA damage checkpoint and G2 arrest in chemo/radio resistant tumors. Radioresistant tumor cells exhibit higher levels of Wnt signaling that leads to upregulation of Myc expression and in turn Chk1/2 expression (*Zhao et al., 2018*). It is plausible that Wnt-dependent upregulation of Chk1 evidenced here and in other contexts could be a general mechanism for the activation of a G2 checkpoint.

The mechanism for G2 arrest in Tr2 tracheoblasts is intimately linked with growth. Tr2 tracheoblasts switch from hypertrophic growth to hyperplastic growth during larval life. The balance between these modes of growth determines tracheal size at the onset of pupariation. We have shown that withdrawal of negative regulators like Wnt/Chk1 leads to a precocious switch from hypertrophy to hyperplasty and to a reduction in tracheal size. An unusual feature of G2 arrest in tracheoblasts is that the concomitant expression of both positive and negative cell cycle regulators. We show here that the juxtaposition of positive and negative regulators is essential for hypertrophic growth and that the withdrawal of either positive or negative regulators perturbs growth. These findings could be broadly relevant to growth control in other systems.

# Materials and methods

## Key resources table

| Reagent type (species) or resource | Designation | Source or reference | Identifiers | Additional information |
|---|---|---|---|---|
| Genetic reagent (*Drosophila melanogaster*) | *btl-GAL4* | *Shiga et al., 1996* | FLYB: FBtp0001208 | This line was a gift from Dr.Shigeo Hayashi |
| Genetic reagent (*Drosophila melanogaster*) | *UAS-Chk1*$^{RNAi}$ | VDRC | 110076 | |
| Genetic reagent (*Drosophila melanogaster*) | *UAS-TCF*$^{RNAi}$ | VDRC | 3014 | |
| Genetic reagent (*Drosophila melanogaster*) | *UAS-Chk1* | *Kizhedathu et al., 2018* and This paper | FLYB: FBgn0261278 | Developed using clone from DGRC: UFO05423 |
| Antibody | Phospho-Chk1 (Ser345) | CST | 2348 | Rabbit monoclonal antibody (1:200) |
| Commercial assay or kit | Tyramide signal amplification system | Thermo Fisher Scientific | B40922 | |

## Fly strains and handling

The following strains were obtained from repositories: *TubGAL80*$^{ts}$; *TM2/TM6b,Tb, UAS-FUCCI, UAS-Wnt2*$^{RNAi}$ (29441), *UAS-Wnt4*$^{RNAi}$ (29442), *UAS-Wnt5*$^{RNAi}$ (28534), *UAS-Wnt6*$^{RNAi}$ (30493), *UAS-WntD*$^{RNAi}$ (28947), *UAS-Wnt10*$^{RNAi}$ (31989), *UAS-CyclinB*$^{RNAi}$ (34544), *UAS-Stg*$^{RNAi}$ (29556), *nkd-lacZ, Chk1-LacZ, fz3-GFP, UAS-ArmS10, UAS-Tkv*$^{QD}$, *Wnt6 KO, Wnt5*$^{400}$, *Wg*$^{ts}$ (Bloomington *Drosophila* Stock Center), *UAS-Chk1*$^{RNAi}$ (110076), *UAS-TCF*$^{RNAi}$ (3014), *UAS-Fz2*$^{RNAi}$ (44391), *UAS-Wg*$^{RNAi}$ (13352), *UAS-Wnt10*$^{RNAi}$ (100867), *UAS-Dsh*$^{RNAi}$ (101525), *UAS-Arm*$^{RNAi}$ (7767), *UAS-Cdc2*$^{RNAi}$ (106130) (Vienna *Drosophila* Resource Center), *UAS-Chk1* (In-house fly facility). The following strains were received as gifts: *btl-GAL4, UAS-ATR, UAS-Wnt10*. Strains were raised on a diet of cornmeal-agar and maintained at 25°C except GAL80$^{ts}$ and *Wg*$^{ts}$ strains that were maintained at 18°C. For

experiments involving GAL80$^{ts}$ and $Wg^{ts}$ strains, the animals were moved to 29°C at indicated stages for indicated time periods. All experiments were performed on animals raised at 25°C unless otherwise indicated.

## Larval staging

Larval staging was performed as previously described (*Guha and Kornberg, 2005*) based on the morphology of the anterior spiracles. L2 larvae were collected and examined to identify animals that had undergone the L2-L3 molt in 8 hr intervals (0–8 h L3). 0–8 h L3 cohorts collected in this method were staged for subsequent time points.

## Immunohistochemistry and imaging

Animals were dissected in PBS and fixed for 30 min with 4% (wt/vol) Paraformaldehyde in PBS. The following antisera were used for Immunohistochemical analysis: Chicken anti-GFP (Aves, 1:500), Rabbit anti-phospho Chk1 (CST, 1:200), Rabbit anti-phospho Smad (CST, 1:150) Rabbit anti-pH3 (Millipore, 1:500), and Alexa 488/568/647-conjugated Donkey anti-Chicken/Rabbit/Mouse secondary antibodies (Invitrogen, 1:200). Tyramide signal amplification was used as per manufacturer recommendations for p-Chk1 detection. The following reagents were used as part of this protocol: Tyramide amplification buffer and Tyramide reagent (Thermofisher), Vectastain A and B and Biotinylated donkey anti Rabbit IgG (1:200, Vector Labs). Tracheal preparations were flat-mounted in ProLong Diamond Antifade Mountant with DAPI (Molecular Probes) and imaged on Zeiss LSM-780 laser-scanning confocal microscopes. Images were processed using Image J. For quantification of cell number, fixed specimens were mounted in ProLong Diamond Antifade Mountant with DAPI and the number of nuclei were counted on an Olympus BX 53 microscope. The DT of the second thoracic metamere was identified morphologically based on the cuticular banding pattern at anterior and posterior junctions.

## Single molecule FISH (smFISH)

Probe sets for *Wg*, *Wnt5*, *Wnt6* and *Wnt10* were designed using Stellaris RNA FISH Probe Designer (Biosearch Technologies, Inc, Petaluma, CA) available online at www.biosearchtech.com/stellarisdesigner. smFISH was performed as per manufacturers instructions. Briefly, the tissues were fixed for 45 min in 4% (wt/vol) paraformaldehyde at room temperature and permeabilized overnight in 70% ethanol. The hybridization was performed overnight at 37°C. The samples were then washed and imaged on a Zeiss LSM-780 laser-scanning confocal microscope. The probes for *Wg* and *Wnt6* were validated by detecting the pattern of the stripe in WL3 wing disc. *Wnt5* probes were validated by staining the trachea from a *Wnt5* null mutant where we did not observe any signal.

## RNA isolation and quantitative PCR

RNA extraction and qPCR were performed as described in *Kizhedathu et al., 2018*. Primer sequences for *Chk1*, *Fz3*, *Wg*, *Wnt5*, *Wnt6*, *Wnt10* and *GAPDH* (internal control) are provided below. Relative mRNA levels were quantified using the formula RE = $2^{-\Delta\Delta Ct}$ method.

The following primer sets were used:

| | |
|---|---|
| *GAPDH* Forward | 5' CGTTCATGCCACCACCGCTA 3' |
| *GAPDH* Reverse | 5' CACGTCCATCACGCCACAA 3' |
| *Chk1* Forward | 5' AACAACAGTAAAACGCGCTGG 3' |
| *Chk1* Reverse | 5' TGCATATCTTTCGGCAGCTC 3' |
| *Wg* Forward | 5' AAATCGTTGATCGAGGCTGC 3' |
| *Wg* Reverse | 5' GGTGCAGGACTCTATCGTTCC 3' |
| *Wnt5* Forward | 5' AGGATAACGTGCAAGTGCCA 3' |
| *Wnt5* Reverse | 5' ACTTCTCGCGCAGATAGTCG 3' |
| *Wnt6* Forward | 5' AGTTTCAATTCCGCAACCGC 3' |

*Continued on next page*

| | |
|---|---|
| *Wnt6* Reverse | 5' TCGGGAATCGCGCATTAAGA 3' |
| *Wnt10* Forward | 5' CACGAATGGCCCGAAAACTG 3' |
| *Wnt10* Reverse | 5' CCCACGGTGCCCTGTATATC 3' |
| *Fz3* Forward | 5' ATGAATGTCGTTCAAAGTGG 3' |
| *Fz3* Reverse | 5' TATAGTAAATGGGGCTTGCG 3' |

## Acknowledgements

We thank Shigeo Hayashi, Anja C Nagel and Prachi Yadav for fly strains and the Central Imaging and Flow Cytometry Facility (CIFF) and Fly Facility at BLiSc for their support. We thank Benny Shilo, members of the Regulation of Cell Fate Theme at inStem and Narmada Khare for comments on the manuscript. Support: Ramalingaswami Fellowship (Department of Biotechnology, Government of India, AG) and institutional funds from inStem (AK, RSK, PV).

## Additional information

### Funding

| Funder | Grant reference number | Author |
|---|---|---|
| Department of Biotechnology, Ministry of Science and Technology | inStem Core Grant | Arjun Guha |
| Department of Biotechnology, Ministry of Science and Technology | InStem Core Grant | Arjun Guha |

The funders had no role in study design, data collection and interpretation, or the decision to submit the work for publication.

### Author contributions

Amrutha Kizhedathu, Conceptualization, Formal analysis, Validation, Investigation, Writing - original draft, Writing - review and editing; Rose Sebastian Kunnappallil, Puja Verma, Investigation; Archit V Bagul, Conceptualization, Methodology; Arjun Guha, Conceptualization, Writing - original draft, Writing - review and editing

### Author ORCIDs

Arjun Guha https://orcid.org/0000-0002-3753-1484

### Decision letter and Author response

Decision letter https://doi.org/10.7554/eLife.57056.sa1
Author response https://doi.org/10.7554/eLife.57056.sa2

## Additional files

### Supplementary files

• Transparent reporting form

### Data availability

All data generated or analysed during this study are included in the manuscript.

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
