## [Decision Letter]

Thank you for submitting your article "Multiple Wnts act synergistically to induce Chk1/Grapes expression and mediate G2 arrest in *Drosophila* tracheoblasts" for consideration by *eLife*. Your article has been reviewed by three peer reviewers, one of whom is a member of our Board of Reviewing Editors, and the evaluation has been overseen by Didier Stainier as the Senior Editor. The following individuals involved in review of your submission have agreed to reveal their identity: Bruce Edgar (Reviewer #2); Greg Beitel (Reviewer #3).

The reviewers have discussed the reviews with one another and the Reviewing Editor has drafted this decision to help you prepare a revised submission.

Summary:

In their manuscript entitled "Multiple Wnts act synergistically to induce Chk1/Grapes expression and mediate G2 arrest in *Drosophila* tracheoblasts," Kizhedathu and colleagues investigate the developmental regulation of Chk1 activation in larval tracheoblasts of the Dorsal Trunk segment Tr2. They find that four Wnt ligands are required to achieve a level of active Chk1 (pChk1) needed to maintain tracheoblasts in G2 arrest. This regulation is achieved by autocrine signaling in which the canonical Wnt pathway is activated by 4 Wnt ligands expressed in the trachea at high levles. Wnt signaling is required for transcription of Chk1. None of the 4 highly expressed Wnts are dispensable. Release from G2 is required for activation of the Dpp/Tkv/pMad pathway that spurs continuing cell divisions.

This is a Research Advance manuscript following up on work reported in a prior publication entitled "Negative regulation of G2-M by ATR/Chk1 (Grapes) facilitates tracheoblast growth and tracheal hypertrophy in *Drosophila*." The current work makes several important contributions. Having initially identified a G2 cell cycle arrest dependent on Chk1 and ATR, but not upon DNA damage, the authors now show that the cell cycle arrest is maintained through a canonical Wnt signal that mediates transcription of Chk1 mRNA. The authors identify 4 Wnt ligands expressed in the tracheoblasts and show that all 4 are required for G2 maintenance via Chk1 transcription, although individually dispensable for *fz3* transcription. The authors also show that the Dpp pathway signal required to drive tracheoblast cell divisions cannot operate during G2 cell cycle arrest. Lastly, authors note that Wnt5 is thought to signal through a nonconical pathway, but in this instance contributes to canonical signaling.

Essential revisions:

1) Additional controls to confirm the requirement for 4 Wnts

– Is there any chance that there are off target effects from the RNAi that may cause this? Do the Wnt ligands impact each other’s expression? Wnt ligand KD followed by qPCR analysis of the Wnt ligands could be useful. This issue is important to discuss, and test.

– Test of second independent RNAi line where classic loss of function alleles are not available.

– Test RNAi of the non-expressed Wnt ligands and addition of a supplemental table documenting the screen (Wnt and other pathways) with RNAi line numbers, drivers, temperatures and results.

2) Clarify ability of overexpressed Wnts to rescue, and determine the requirement for nonconical Wnt pathway

– Address whether derailed or doughnut are required in tracheoblasts.

– Authors Wnt threshold model hinges on ability of overexpressed Wnt to compensate for loss of one Wnt ligand. However, this was only reported for one loss of function case (Wnt 6 RNAi), and only with one overexpressed Wnt (Wnt5). Authors should test ability of other Wnts to rescue and should also address whether a conventional Wnt can substitute, when overexpressed, for Wnt5.

3) Address inhibition of mitoses during L2:

– The authors show that downregulating the Wnt pathway in L2 stage does lead to the reduction of Chk1 mRNA (Figures 3A and 2, respectively), but that this does not result in tracheoblast mitoses. This indicates that in L2, in contrast to L3, there must be some additional control which is lifted after L2/L3 metamorphosis. The authors should discuss this issue and present possible explanations. If they have any more relevant data, this should also be presented.

4) Move Figure 5—figure supplement 1 into Results.

– Discussion paragraph four. Minimally, it is not appropriate to introduce data in the discussion that is not discussed in the main Results section. Moreover, the experiments and results are super interesting. I could be wrong because I am not an expert in the cell cycle, but I don't think the idea of a G2 arrested cell continuing to grow physically because it still expresses cell cycle promoting genes while in G2 arrest is really out there in the literature on cell and organ size control (although I think mammalian oocytes arrest in G2 and they get really big, but they also may have bridges to nurse-like cells to supplement growth). As such, I would strongly encourage the authors to make Figure 5—figure supplement 1 part of the main figures and discuss the basic findings in the Results section. Then it could be revisited in the Discussion section to put the result in the bigger picture context.

---

## [Author Response]

Essential revisions:1) Additional controls to confirm the requirement for 4 Wnts– Is there any chance that there are off target effects from the RNAi that may cause this?

We have used the UP-TORR (Updated Targets of RNAi Reagents) to analyse the potential off target effects of the Wnt ligand RNAi lines used in the manuscript. It provides up-to-date annotation of the RNAi lines that can be sourced from public collections. The analysis shows that the Wnt RNAi lines that we have used are indeed specific to their respective Wnts and are unlikely to have off-target effects (OTE, see Author response table 1).

Author response table 1

OTE: off target effect on other Wnts

Yanhui Hu, Charles Roesel, Ian Flockhart, Lizabeth Perkins, Norbert Perrimon, and Stephanie E Mohr. 2013. “UP-TORR: online tool for accurate and Up-to-Date annotation of RNAi Reagents.” Genetics, 195, 1, Pp. 37-45.

Do the Wnt ligands impact each other’s expression? Wnt ligand KD followed by qPCR analysis of the Wnt ligands could be useful. This issue is important to discuss, and test.

We have performed qPCR experiments to determine how knocking down expression of one Wnt ligand impacts the expression of other ligands. (Figure 4—figure supplement 1D). We find that a reduction in the expression of one ligand (*Wg, Wnt5, Wnt6, Wnt10*) does not impact the expression of the remaining ligands. Based on this data we infer that expression of Wnts are not dependent on each other and that each Wnt contributes independently toward G2 arrest.

– Test of second independent RNAi line where classic loss of function alleles are not available.

We sourced an independent RNAi line for *Wnt10* from VDRC to test the role of Wnt 10 in G2 arrest. We find that expression of *Wnt10^RNAi^* also it leads to precocious proliferation at 16-24 h L3. This data further supports the finding that *Wnt10* is required for G2 arrest (Figure 4—figure supplement 1A).

– Test RNAi of the non-expressed Wnt ligands and addition of a supplemental table documenting the screen (Wnt and other pathways) with RNAi line numbers, drivers, temperatures and results.

The supplemental table is provided in Figure 4—figure supplement 2. Precise descriptions of the Wnt RNAi lines that have been used is now included in the Materials and methods section.

2) Clarify ability of overexpressed Wnts to rescue, and determine the requirement for nonconical Wnt pathway– Address whether derailed or doughnut are required in tracheoblasts.

We have shown that it is possible to rescue the *Btl-Wnt6^RNAi^* phenotype by overexpression of *Wnt5*. We now show that it is also possible to rescue the *Btl-Wnt6^RNAi^* phenotype by overexpression of *Wnt10*. This data has been included in the text. Based on the results of *Wnt 5/10* overexpression we infer that the Wnts are able to complement each other. The overexpression of *Wg* could not be tested as it is lethal.

Our finding that *Btl-Wnt 5^RNAi^* abrogates G2 arrest suggests non-canonical Wnt signalling could also play a role in mediating arrest. *Wnt5* activates the non-canonical pathway by binding to the RYK family receptor tyrosine kinases Derailed and Doughnut. We have examined how knockdown of Derailed *(Btl-Gal4/UAS-Drl^RNAi^)* and Doughnut *(Btl-Gal4/+; UAS-Dnt^RNAi^/+)* in the trachea impacts G2 arrest (see graph in Author response image 1). Neither the knockdown of Derailed nor Doughnut leads to precocious mitotic entry. While this does not eliminate the possibility that the non-canonical Wnt signalling pathway has a role in mediating G2 arrest, we have found no evidence for this.

**Author response image 1. sa2fig1:** 

– Authors Wnt threshold model hinges on ability of overexpressed Wnt to compensate for loss of one Wnt ligand. However, this was only reported for one loss of function case (Wnt 6 RNAi), and only with one overexpressed Wnt (Wnt5). Authors should test ability of other Wnts to rescue and should also address whether a conventional Wnt can substitute, when overexpressed, for Wnt5.

As stated above, we have tested whether *Wnt10* overexpression can rescue the Btl-Wn6 RNAi phenotype. We find that the overexpression of *Wnt10* can also rescue the loss of Wnt6. Together with the data that *Wnt5* overexpression can also rescue the *Btl-Wnt6^RNAi^* phenotype, these findings suggest that one Wnt can compensate for another.

3) Address inhibition of mitoses during L2:– The authors show that downregulating the Wnt pathway in L2 stage does lead to the reduction of Chk1 mRNA (Figures 3A and 2, respectively), but that this does not result in tracheoblast mitoses. This indicates that in L2, in contrast to L3, there must be some additional control which is lifted after L2/L3 metamorphosis. The authors should discuss this issue and present possible explanations. If they have any more relevant data, this should also be presented.

There is indeed an additional mechanism of negative control that is lifted after the L2/L3 molt. In our earlier paper (Kizhedathu et al., 2018) we showed that induction of a constitutively active Cdc2 (Cdc2^AF^) cannot induce mitoses in L2 but can after the L2/L3 molt. We have reason to think that this additional layer of control could be linked to mitochondrial dynamics. We find that the mitochondria are fused in L2 and fragment immediately after the molt. We are currently investigating how mitochondrial dynamics are regulated and how these dynamics impact the cell cycle program. These findings are beyond the scope of this manuscript.

4) Move Figure 5—figure supplement 1 into Results.– Discussion paragraph four. Minimally, it is not appropriate to introduce data in the discussion that is not discussed in the main Results section. Moreover, the experiments and results are super interesting. I could be wrong because I am not an expert in the cell cycle, but I don't think the idea of a G2 arrested cell continuing to grow physically because it still expresses cell cycle promoting genes while in G2 arrest is really out there in the literature on cell and organ size control (although I think mammalian oocytes arrest in G2 and they get really big, but they also may have bridges to nurse-like cells to supplement growth). As such, I would strongly encourage the authors to make Figure 5—figure supplement 1 part of the main figures and discuss the basic findings in the Results section. Then it could be revisited in the Discussion section to put the result in the bigger picture context.

Done.